# The Effect of Age and Sampling Site on the Outcome of *Staphylococcus aureus* Infection in a Rabbit (*Oryctolagus cuniculus*) Farm in Italy

**DOI:** 10.3390/ani10050774

**Published:** 2020-04-29

**Authors:** Anna-Rita Attili, Patrizia Nebbia, Alessandro Bellato, Livio Galosi, Cristiano Papeschi, Giacomo Rossi, Martina Linardi, Eleonora Fileni, Vincenzo Cuteri, Francesco Chiesa, Patrizia Robino

**Affiliations:** 1School of Biosciences and Veterinary medicine, University of Camerino, Via Circonvallazione 93/95, 62024 Matelica (MC), Italy; livio.galosi@unicam.it (L.G.); giacomo.rossi@unicam.it (G.R.); marti.linardi@gmail.com (M.L.); eleonora.fileni@studenti.unicam.it (E.F.); vincenzo.cuteri@unicam.it (V.C.); 2Department of Veterinary Sciences, University of Turin, Largo P. Braccini 2, 10095 Grugliasco (TO), Italy; patrizia.nebbia@unito.it (P.N.); alessandro.bellato@unito.it (A.B.); francesco.chiesa@unito.it (F.C.); patrizia.robino@unito.it (P.R.); 3Interdepartmental Animal Facility, University of Tuscia, Largo dell’Università snc, 01100 Viterbo (VT), Italy; papeschivet@gmail.com

**Keywords:** methicillin-sensitive *Staphylococcus aureus* (MSSA), spa-typing, rabbits, lesion, nasal carriage, Italy

## Abstract

**Simple Summary:**

*Staphylococcus aureus* contamination has been described in food-producing animals and farm workers involved in the primary industrial production of rabbits, pigs, cattle and poultry. This study describes the effects of age and colonization of body on *S. aureus* detection from rabbits raised intensively for meat production, and evaluates these parameters as possible risk factors for lesions by *S. aureus*. In addition, a genotypic characterization was performed for 96 *S. aureus* strains out of 595 that were isolated, including three from farm workers. It was observed that the risk of skin lesions increased with the number of colonized body sites and with age. All isolates were low-virulence methicillin-sensitive-*Staphylococcus aureus* (LV-MSSA). We found five different clonal lineages: spa-t2802, the most frequently detected (53.1%), also in all human samples; spa-t491, the second most detected (38.5%); spa-t094; t605; and spa-t2036. The same spa-type was observed in both animal and worker samples, showing that *S. aureus* strains could easily circulate in a community. The two most frequent strains were associated with noses, but not with age categories nor with the number of colonized sites. The circulation of LV-MSSA should not be underestimated, as they could determine damage or even acquire and spread resistance genes.

**Abstract:**

A study in an intensive Italian rabbit farm was carried out to assess the effect of age category and anatomical site on detection of *S. aureus* and to describe the diversity of spa-types within farm, including workers. On 400 rabbits of different age, 2066 samples from the ear, nose, axilla, groin, perineum and lesions were analyzed and 595 *S. aureus* were isolated. In total, 284 (71.0%) were colonized by *S. aureus* in at least one anatomical site and 35 animals (8.8%) had lesions. *S. aureus* prevalence was higher in adults than other age categories. Ear (29.4%) and nose (27.2%) were the most frequently colonized sites. The presence of lesions due to *S. aureus* was strongly associated with the colonization of at least one body site and the risk of lesions was proportionate to the number of sites colonized by *S. aureus*. In addition, a genotypic characterization was performed for 96 *S. aureus* strains randomly selected. All isolates resulted as low-virulence methicillin-sensitive-*Staphylococcus aureus* (LV-MSSA). Five different clonal lineages were found: spa-t2802, the most frequently detected (53.1%), also in human samples; spa-t491 (38.5%); spa-t094; t605; and spa-t2036. Strains t094, t491 and t2036 have not been isolated in Italy before.

## 1. Introduction

In recent decades, the epidemiology of *Staphylococcus aureus* infections has received growing attention because of their importance in veterinary medicine, and the increased evidence of their zoonotic potential. *S. aureus* is an extremely flexible organism: it can be a commensal but also a dangerous and easily adaptable pathogen with high persistence and multiplication capacity in a wide variety of environments and conditions, able to cause a wide range of diseases in both humans and animals [1,2,3,4,5]. Distinct clones have been identified within the global population of *S. aureus* that associate closely with specific hosts [6,7,8]. Staphylococcal infections cause considerable economic losses in all farming conditions, but particularly in intensive farming, because of poor production performances, infertility, death, and the increased amount of slaughtering [9], with a prevalence ranging from 40% [10] to 70% [11]. In domestic rabbits (*Oryctolagus cuniculus*), staphylococcal infections usually result in skin lesions that often evolve involving subcutaneous tissues, but they can also generate more complex and varied secondary lesions such as pododermatitis [12,13], multisystemic abscesses (with internal organ abscesses sporadically observed), and mastitis [14,15,16]. Successful infection depends on the bacterium–host interaction and on virulence factors produced by *S. aureus*. In rabbits, two principal clinical types of *S. aureus* infections could be distinguished. Low virulence (LV) strains determine infections that remain limited to a small number of animals, and only have little economic importance; high virulence (HV) strains spread throughout the flock, leading to chronic problems and subsequent declines in production performances [17].

The discovery of emergent methicillin-resistant *Staphylococcus aureus* (MRSA) in farmed rabbits [18,19] and companion rabbits [20,21,22], including strains primarily associated to human hospitals and then found in livestock and farmers (e.g., livestock associated (LA) clonal complex 398 MRSA [20]), has raised concerns about the *S. aureus* population in rabbits [23]. These staphylococci are resistant to most beta-lactams antibiotics and are a serious problem of public health.

The aims of this study were as follows: (i) to observe the age and sampling site effect on the outcome of *S. aureus* colonization in rabbits; (ii) to assess the prevalence of lesions due to *S. aureus* infection; and (iii) to characterize the main clonal lineages of animal and human strains and relative closeness among them.

## 2. Materials and Methods

### 2.1. Animals and Breeding

The study was carried out on a commercial medium-sized intensive rabbit breeding farm in Central Italy. It consisted in a large shed where fattening, breeding, and replacement rabbits were all in the same shed. Animals were raised in overlapping cages and the “battery” method on parallel rows separated by corridors. At the time of the research, about 8000 animals, including 1700 breeding rabbits, were present. The reproduction was by artificial insemination, and about 70% and 30% were the internal and external replacement rabbits, respectively. All animals were submitted to a half-yearly MEV vaccination plan and the average annual production was around 40–45,000 slaughter rabbits between 82 and 90 days of age. The average annual mortality of 8–9% in the nest, and 7–8% in fattening, were recorded. Prior to beginning of the study, the rabbits were kept under standard conditions of animal housing, with good management and cleaning of animal cages and watering holes. They had light and dark conditions for each 12 h and temperature of 20–24 °C. The feeding consisted of a single cycle feed with addition of barley, finishing weaning at 35 days of age.

### 2.2. Sampling

In June 2018, a total of 400 rabbits, represented 5% of the total animals reared, were randomly selected: 26 young (40–45 days of age), 312 adults (60–80 days of age), 27 breeding rabbits, and 35 replacement rabbits.

Skin swabs from five anatomical sites, nose (N), ear (E), axilla (A), groin (I), and perineum (P), were collected (n = 2000) from each animal. When skin lesions (L) were present, additional swabs (n = 66) were sampled (n = 5 from young, n = 39 from adult, n = 20 from breeding rabbits, and n = 2 from replacement rabbits). Lesions were circumscribed, ulcerate skin areas with purulent exudate, in various body sites (ear, eye, nose, dewlap, tarsus, skin of the interdigital spaces).

Each swab was placed into transport medium (Amies, Liofilchem**^®^**, Roseto degli Abruzzi, Italy) and kept at 4 °C until delivery to the microbiological laboratory, which occurred within 48 h. Animals treated with antibiotics within 30 days were excluded from the study. Nasal and hand skin swabs were also collected from three farm workers, after obtaining their informed consent.

### 2.3. Isolation and Identification of Staphylococcus Aureus

For bacteriological culture, each swab (n = 2066) was enriched with Triptic Soy Broth (Liofilchem**^®^**, Roseto degli Abruzzi, Italy) for 6 h at 37 °C and was then inoculated onto nutrient and selective media as Columbia agar (Liofilchem**^®^**, Italy) supplemented with 5% ovine blood, Baird-Parker agar (Liofilchem**^®^**, Italy), Mannitol Salt Agar (Liofilchem**^®^**, Italy) and Chromatic™ *Staphylococcus aureus* (Liofilchem**^®^**, Italy). After 24–48 h of incubation at 37 °C, presumptive *S. aureus* colonies were studied according to usual bacteriological standards (Gram staining, catalase and coagulase tests) and biochemical identification was carried out by a commercial kit (RapID STAPH, Remel Oxoid, Italy). Then, *S. aureus* strains were stored at −80 °C (CryoBank, MAST Diagnostic, Reinfeld, Germany) for molecular analysis.

Out of all of the isolates, 96 *S. aureus* strains were randomly selected for molecular characterization. In particular, 72 *S. aureus* were strains isolated from healthy sites of the four age animal categories, 21 strains were from animal skin lesions, and three strains were from farm workers.

### 2.4. Genotypic Characterization

DNA extraction was performed using InstageneTM Matrix (Biorad**^®^**, Segrate, Italy). Purified DNA was quantified with Thermo Fisher Nanodrop and stored at −80 °C until use. To ascertain *S. aureus* and MRSA identification, multiple polymerase chain reactions (PCRs) were performed targeting *S. aureus* specific thermo-nuclease gene (*nuc*) and methicillin-resistance (*mecA* and *mecC* homolog, originally designated *mec_ALGA251_*) genes, as described previously [24,25].

To differentiate between high virulence (HV) and low virulence (LV) *S. aureus* strains, and to identify atypical highly virulent strains [26], a multiplex PCR method [27] was used for the detection of *bbp*, *selm* and *flank* genes.

The spa-typing method was performed to amplify Staphylococcal Protein A (*spa*) gene [25,28]. Sequences were analyzed using MEGA Software, and spa-types were identified according to Ridom SpaServer data with Fortinbras Spatyper [28,29]. Cluster analysis to assign spa-clonal complexes (spa-CCs) to spa-types was performed by applying the based upon repeated pattern (BURP) algorithm with Bionumerics Seven (©Applied Maths NV, Sint-Martens-Latem, Belgium), and using other sequences of spa-types recreated by hand in FASTA format. The BURP algorithm with default parameters (exclusion of spa-types shorter than five repeats and clustering of spa-types if cost was lower than or equal to 4) was applied.

The spa-type sequences found were submitted to GenBank as “Immunoglobulin G binding protein A of *S. aureus*, partial coding sequences” (Accession numbers: IT281RE; IT007RL; IT014RI; IT017RN; IT001HN).

### 2.5. Statistical Analysis

Statistical analysis was performed by software STATA version 13.0 (©StataCorp LLC., College Station, TX, USA), using χ^2^ test to evaluate the significance of differences between qualitative variables, and Fisher’s exact test to deal with empty cells. Odds ratios (ORs) were used as measures of association between outcome and risk factors, with 95% confidence intervals obtained by the Woolf’s method for crude ORs. Mantel-Haenszel OR was used to control for confounding factors, with 95% support intervals obtained by maximum likelihood method (ML). Logistic regression models, chosen by forward selection method, were performed to evaluate the effect on outcome of discrete quantitative and binomial variables as risk factors and confounders, and to predict expected prevalence of lesions by *S. aureus*. Particularly, the outcome of interest was defined as lesion from which *S. aureus* was isolated. As possible risk factors for lesions by *S. aureus*, colonization of body sites, nasal carriage, age category and bacterial lineages (spa-types), were evaluated. Association measures were reported as OR, with 95% support intervals obtained by ML. An association plot was produced using package *vcd* of software R version 3.6.1 [30].

## 3. Results

### 3.1. Sample Collection and Bacterial Isolation

In total, 2066 skin samples were collected from 400 animals. The sampling distribution and prevalence of *S. aureus* in relation to the age animal category and the anatomical site are shown in Table 1. Out of 2000 swabs sampled from the five health anatomical sites (nose, ear, axilla, groin, perineum), 552 swabs were positive for *S. aureus* coming from 284 rabbits (71.0%, n = 400), and at least in one anatomical site. Between the different sites, a significant difference in the frequency of isolation was recorded (*p* < 0.05), while a similar and high isolation was observed in both the auricular (29.4%) and nasal swabs (27.2%, n = 592; χ^2^ = 0.01, *p* = 0.925).

Out of 284 rabbits, 174 (61.3%) had more than one site colonized, distributed as follows: 105 rabbits (37.0%) had two; 49 (17.2%) had three; 15 (5.3%) had four; and in 5 (1.8%) animals all five sampled anatomical sites were positive for *S. aureus* colonization. A significant difference was recorded for all sites across all age categories (*p* < 0.05), except between young rabbits (4.6%) and the replacement rabbits (3.9%; χ^2^ = 0.07, *p* = 0.785).

During sampling, 66 lesions were observed in 58 animals: Only one lesion in 51 (87.9%, n = 58) rabbits; two lesions in six (10.3%) rabbits; and three in one (1.7%) rabbit. Out of those, 35 (8.8%) animals presented lesions by *S. aureus*, and of these, 30 (7.5%) had only one lesion, while five rabbits (1.3%) had two lesions.

Out of 66 lesions, *Staphylococcus* spp. were isolated from 54 (81.8%), while *S. aureus* was identified in 40 of them (74.1%), and it was significantly associated with lesions (χ^2^ = 34.0, *p* < 0.000). In relation to the age category, *S. aureus* was isolated in 20%, 84.6%, 30%, and 0% of swabs collected from lesions present in young (n = 5), adult (n = 39), breeding (n = 20), and replacement rabbits (n = 2), respectively (Table 1).

The presence of lesions due to *S. aureus* was strongly associated with the colonization of at least one body site (OR = 7.5, 95% CI: 1.8–31.8). The higher number of colonized body sites was associated with an increased risk of lesions due to *S. aureus* (Table 2) suggesting that when the animal was colonized in more than one body site, the probability that the lesion is caused by *S. aureus* is higher. The estimated increase in risk due to each more site colonized was OR = 1.7, 95% CI: 1.3–2.3. There was a positive trend of crude risk from OR 4.5 for only one colonized site to OR 14.3 for four or more sites (Table 2).

Prevalence of colonization differed among anatomical sites, particularly in those that were positive for *S. aureus*: 174 in the ears; 63 in axillae; 161 in noses; and 77 perinea and inguinal regions (Table 1). Some of those were significantly associated with lesions by *S. aureus*: axilla (χ^2^ = 7.1, *p* = 0.008); nose (χ^2^ = 10.3, *p* = 0.001); inguinal region (χ^2^ = 5.6, *p* = 0.018). Nose colonization had a higher correlation with colonization than the other sites (Spearman’s rho ranging from 0.129 to 0.216), and therefore swabs from the nose were analyzed more in detail. Although all sites were positively associated to the outcome, nose colonization was significantly associated with increased risk of lesions by *S. aureus* (OR = 2.4, 95% CI: 1.1–5.2).

Rabbits were unevenly distributed among age categories. When the analysis was carried out to understand if lesions by *S. aureus* were more representative in a particular age category or in animals with more than one colonized body site, a weak association was observed between categories and lesions by *S. aureus* (Fisher’s exact *p* = 0.048), and between categories and colonization by *S. aureus* of at least one body site (χ^2^ = 10.1, *p* = 0.018). In young rabbits, the most contaminated site by *S. aureus* was the ear (33.3% isolates), with no significant difference towards the prevalence recorded in the other anatomical sites (*p* > 0.05). In particular, all young and replacement rabbits had none or few sites colonized by *S. aureus*, and never were there more than three. While in young rabbits only in 20% (n = 5) of the lesions *S. aureus* was cultured, in replacement rabbits no lesions due to *S. aureus* were recorded. Among adult fattening rabbits, out of 478 isolates, the nose (29.1%) and the ear (27.6%) were the most contaminated sites (Table 1). Although adult rabbits showed up to five colonized sites, most of them had few sites colonized, while breeding rabbits had in general more colonized sites than other categories and were the only category to be significantly associated to a higher number of colonized sites (χ^2^ = 20.7, *p* = 0.001). Compared to young rabbits, adult fattening and breeding rabbits had a higher risk of developing lesions by *S. aureus*, OR = 2.6 (95% CI: 0.3–19.6), and OR = 5.7 (95% CI: 0.6–52.4), respectively. Statistical data evidenced that age category cannot explain all the differences regarding presence of lesions, and other factors, such as the number of colonized sites, seem to play a role. Breeding rabbits had a higher probability to become colonized by *S. aureus* in at least one body site (OR = 4.2, 95% CI: 1.1–15.7), showing positive correlations (Spearman’s rho ranging from 0.071 to 0.126), and a positive association with the colonization of all sites, in particular the ear (OR = 3.8, 95% CI: 1.2–11.8) and nose (OR = 6.9, 95% CI: 1.9–25.4).

Nasal carriage alone was evaluated as a potential risk factor for developing lesions by *S. aureus*, since it was strongly associated with them (χ^2^ = 10.3, *p* = 0.001), and crude risk of OR = 3.2 (95% CI: 1.5–6.5) was observed. The risk from nasal carriers was due only to adult fattening and breeding rabbits, as young and replacement rabbits added no information. Age-stratified odds ratios showed that the nasal carriage effect varied widely among different age categories, being OR = 2.6 (95% CI: 1.2–5.8) for adult and OR = 4.0 (95% CI: 0.4–41.8) for breeding rabbits.

The statistical analysis showed that the number of sites colonized by *S. aureus* was the stronger risk factor observed for developing lesions, with a dose–response pattern that reinforces the plausibility of our findings (Figure 1).

Out of a total of 592 isolates, (healthy skin, n = 552; lesions, n = 40) a sample of 96 *S. aureus* was selected for genotypic characterization. The sample was distributed as follows: 72 strains representing the four breeding categories (n = 2, 2.8% of *S. aureus* infection in young rabbits; n = 52, 72.2% in adult rabbits; n = 14, 19.4% in breeding rabbits; n = 4, 5.5% in replacement rabbits); 21 strains were isolated from lesions; and three strains came from farm workers. Most of the samples were cultures from nose swabs (n = 64), due to the high correlation found between the colonization of this site with other anatomical regions, and for the spreading role of nasal carriage of *S. aureus*, because rabbits—similar to many other animals—use this part to reach all other body sites. Paired samples (n = 16) were selected from rabbits with both lesions and nose or ear colonization; some samples (n = 2) were from lesions in rabbits with no colonized sites, while the others were chosen to respect age category and level of colonization distributions.

### 3.2. Genotypic Characterization

All 96 *S. aureus* strains were negative for the presence of *bbp*, *selm*, *flank* genes of virulence and for *mecA* and *mecC* gene detection; therefore, they were classified as LV-MSSA strains.

The *spa* gene was found in all tested isolates (n = 96). We identified 5 different spa-types: t094, t491, t2036, t2802, and t605. A bimodal distribution was identified, with the most frequently spotted spa-type, t2802 (53.1%), also detected in all human samples, and t491 being the next most frequently spotted spa-type (38.5%) (Table 3).

Due to its shortness, t605 could not be clustered in any clonal complex. The other four spa-types were arranged into three spa-CCs using BURP algorithm: spa-CC267, to which belongs t2802; spa-CC084, to which belongs t094 and t491; and spa-CC012, with t2036. The last two clustered together in our study, due to the relative closeness (distance < 4) among the spa-types found.

Figure 2 shows clusters and the distances between spa-types.

Since spa-type t094 was found only in replacement rabbits, t605 in adult, and t2036 in breeding rabbits, they were significantly associated to age categories (Fisher’s exact *p* = 0.005), but their scarce frequencies did not facilitate a generalization of our findings to a larger population. The two most represented spa-types, t491 (n = 37) and t2802 (n = 51), were not associated with animal categories (Fisher’s exact *p* = 0.192), but t2802 is the only one that has been found in farmers (n = 3).

Spa-types were not associated with the number of colonized sites (Fisher’s exact *p* = 0.526). Out of 11 isolates from the auricular region, five were t2802, three were t094, two were t2036, and one was t491; meanwhile, out of 64 isolates from the nose, 39 were t2802, 24 were t491, and one was t094. An overall association with sampling sites was found, with the most represented spa-types being isolated mainly from nose (Fisher’s exact *p* < 0.000). Figure 3 shows associations between spa-types, sampling sites and age category.

Spa-type t605 isolates (n = 2) were drawn from lesions, while t094 and t2036 only came from healthy skin swabs. Spa-type t491 and t2802 isolates were drawn from lesions 12 out of 37 and 7 out of 51 times, respectively. t491 was positively associated to lesions by *S. aureus* (crude OR = 3.9, 95% CI: 1.3–11.3), even controlling for both age category and number of colonized sites (OR = 4.1, 95% CI: 1.2–13.2).

Out of the eight rabbits in which both isolates from lesions and nose were analyzed, six had the same spa-type, and two had different spa-types. Particularly, when lesions differed from the nose, t2802 in the nose and t491 or t605 in lesions were observed; when they had the same spa-type in both sites, t2802 was isolated most frequently (66.7%, n = 6). In lesions from rabbits which had no other sites colonized by *S. aureus*, t491 and t605 were isolated once each.

## 4. Discussion

In this study, the overall prevalence of rabbits colonized by *S. aureus* in at least one body site was quite high (71.0%), although being within the same range observed by Hermans et al. [10] and Agnoletti et al. [11], who reported a *S. aureus* prevalence of 60.5% in Belgian rabbit farms and 77.6% in Italian farms, respectively. Rabbits were colonized mainly in the nose, which suggests that it was a stable colonization in balance between bacterium and host. The high colonization rate (71.0%) is not only attributable to the intrinsic capacity of certain clones, but also to the crowding of the animals, their frequent manipulation, and the top-down distribution of the pure lines of parents, which favors the dissemination of *S. aureus* clones within the groups. These phenomena are intrinsic characteristics of the industrial breeding rabbit system rather than the *S. aureus* clone of interest.

All strains, including those isolated from skin lesions, resulted with LV-MSSA, suggesting that the *mecA* and *mecC* gene presence is still not a widespread problem in this type of breeding in our country, although MRSA was isolated in commercial rabbits in 2014 [19,31].

The limited economic impact of low virulence *S. aureus* (LVSA) leads the farmer to control it rather than spending resources to prevent it; consequently, little information is available on the role of those bacteria in lowering productivity of the animals. During a clinical experiment, Meulemans et al. [23] demonstrated that wounds infected by LVSA evolve to abscesses and severe lesions, only to heal after about 14 days. Although the rabbit does not die, it suffers a slowly evolving disease that could reduce its growing and productive performances or determine depreciation or condemnations at slaughtering.

As reported previously, *S. aureus* can be isolated from the whole body of rabbits: The nose, ear, skin of the interdigital spaces, the skin of the forelegs, the skin of the axillary and inguinal regions, the skin around the nipples, the perineum, the vagina, and the foreskin [10,32,33,34,35,36]. The nose was considered the best choice here, due to the homogeneous distribution between categories. Moreover, nasal colonization was positively related with the colonization of all other sites, confirming that the nostrils are the best place to isolate *S. aureus* from rabbits. The ethology indicates that rabbits spend an elevated amount of time washing themselves, eating and performing caecotrophy, meaning that they can reach virtually every part of the body with their nose; thus, the nose should represent the colonizing population of all body sites. As Hermans et al. [14] stated, our results confirmed that more than only one body site should be tested in order to identify the presence of *S. aureus*, and that sampling only the nose could result in a loss of information.

Adult and breeding rabbits have many differences to young and replacement rabbits, in terms of physiology, weight, diet, time spent in the same environment, type of cage, and behavior. Age category represented a summary measure to consider all of them, and since it was associated in different ways with lesions and colonization by *S. aureus*, it was not appropriate to give a summary result for all the rabbits, but it was necessary to consider them separately. It is presumable that the greatest number of lesions observed on adult animals derived from the type of cage and the hyper energetic diet, while for breeding rabbits the behavior during mating could induce wounds that could become infected. However, those reasons were not enough to explain the excess of lesions observed, and so the number of infected sites, or the level of body colonization by *S. aureus* should be considered. Not only was this associated with a higher probability of lesions, but it showed a dose–response effect that also remained, considering the differences related to age categories. On the other hand, nasal carriage, which was higher among adult and breeding rabbits, could partially explain the excess of lesions and suggests a stable bacterium–host interaction. The relatively low prevalence of severe lesions (8.8%) compared to colonization of at least one body site suggests that most of the animals were asymptomatic carriers and that the infection was due to perturbation of this equilibrium. What we observed in this study could reflect the hypothesis that all rabbits, irrespective of age group or production category, the secondary lesions by *S. aureus* derive from abscessualization of primitive traumatic lesions (insect bites, repeated scratching, “cage” sores in the tarsi), and the risk increases when more sites are involved [37].

Spa-typing is a fast and powerful tool for epidemiology research, since it allows one to identify the predominant lineages circulating in the flock, and to trace them back to clonal complexes. In this farm, five different spa-types were identified and t491 and t2802 were the most frequently detected. Among rabbit categories, the bimodal distribution of spa-types, namely t2802 and t491, partially reflects the spatial allocation of the subjects within the farm. Spa-type t094 was isolated only from replacement rabbits, and from them all, while t2036 was observed only from breeding rabbits, corroborating the belief that subjects purchased from other farms for breeding purposes could act as carriers of new, and potentially pathogenic, strains.

All human *S. aureus* strains were identified as t2802, the most frequently represented in animal isolates, thus corroborating the belief that the transmission of *S. aureus* strains, both HV and LV, between rabbits or from man to rabbits, can take place directly or indirectly, by contact from cage to cage, air, or through the hair and feed [14,26,38]. Since 2004, t2802 was reported as the cause of hospital outbreaks and nosocomial infections in humans [34], and it has been isolated in Italy as recently as 2018, but its isolation remains quite rare. As reported by Ridom SpaServer [28], a free access database that keeps up to date the register of spa-types occurrence, t094, t491 and t2036 have not been isolated in Italy before. Spa-types t605 and t491, despite not having the virulence factors, were positively associated with lesions, but their role was presumed to be as opportunistic rather than pathogen strains. There was no strong indication that spa-types were selectively associated with anatomic sites; however, it was mainly in the nose that the most frequent spa-type was found, while other spa-types were associated with other sites. This could be due to the stable presence in the flock of those strains, and their prevalence was so high that it could be dismissed that there were other more represented clones. Conversely, the same spa-types were observed both in animal and worker samples, showing that *S. aureus* strains could easily circulate in a community. Since spa clonal complexes could be related to MLST clonal complexes with good approximation [39], the t2802, which belongs to spa-CC267, a cluster mainly constituted by LA-MSSA, could be referred to as CC97, a clonal complex often described in human medicine for nosocomial infections [40,41]. Spa-types t491, t094 and t2036, although being part of different spa-CCs, cluster together in spa-CC084, which is reportedly related to CC15, again a complex linked to hospitals and humans [40,41]. Grouping types into clonal complexes is essential for monitoring the distribution and evolution of clones. Although no inference could be made about the evolution of the strains and their origin, those findings suggest that boundaries between community–acquired and livestock–acquired are blurred, and since in microbiology there is little that can be considered steady, keeping a free-access database up to date is essential to monitor the epidemiologic situation.

This study cannot claim to be representative of the Italian rabbit population, but it does represent an initial step towards understanding the main risk factors that could increase infections by *S. aureus*.

## 5. Conclusions

Among rabbit breeding farms, staphylococcosis represents one of the main problems due to its economic impact. *S. aureus*, with its opportunistic behavior, can survive on the skin of animals and humans for a long time, making them agents of the infection.

This study describes the effects of age and body colonization on detection of *S. aureus* from rabbits reared intensively for meat production in a standard Italian breeding farm, and shows that those are significantly associated with the risk of skin lesions. Particularly, the *S. aureus* occurrence is higher in adult rabbits than other categories and ear and nose are the most frequently sites colonized. The presence of lesions due to *S. aureus* is strongly associated with the colonization of at least one body site, and the risk of lesions due to *S. aureus* is proportionate to the number of sites colonized by *S. aureus*.

The authors observe that the nasal carriage is significantly associated with an increased risk of lesions due to *S. aureus*, and an association with nose colonization was found with the most represented clonal lineages in the rabbit breeding farm, but no association between spa-types and the number of colonized sites was found. The same spa-types were observed in both animal and worker samples, showing that *S. aureus* strains could easily circulate in a closed community with the workers-animal interaction.

Finally, the authors report a circulation only of LV-MSSA strains that should not be underestimated, as they could nonetheless cause economic damage, or even acquire and spread resistant genes. Few studies have been published on low virulence *S. aureus* in rabbits, and little is known about their prevalence. Our study focused on the impact of those bacteria on rabbit breeding activity, using a simple yet powerful epidemiologic approach to highlight the main risk factors.

## Figures and Tables

**Figure 1 animals-10-00774-f001:**
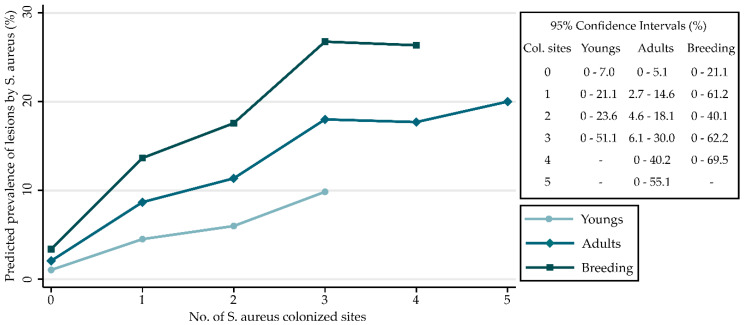
Expected prevalence of *S. aureus* lesions: Predicted prevalence of lesions due to *S. aureus* based on the number of colonized anatomical sites and age categories. Confidence intervals are shown in the table.

**Figure 2 animals-10-00774-f002:**
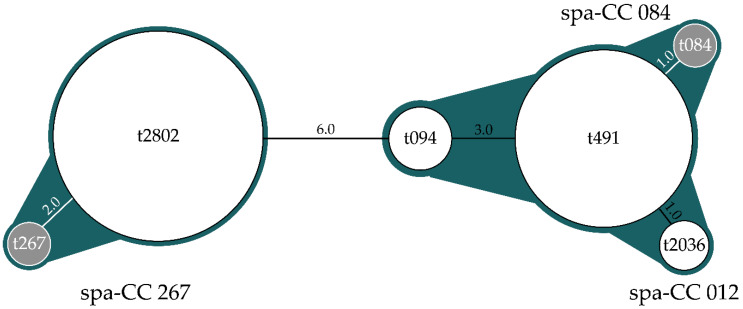
Relationship between clones: clusters are highlighted in blue, with *spa*-CCs notations. Each circle’s size is proportional to spa-type frequencies and distance between them is reported. Spa-CCs in grey are the founders of the clonal complexes. Distances between spa-types were analyzed by based upon repeated pattern (BURP) algorithm.

**Figure 3 animals-10-00774-f003:**
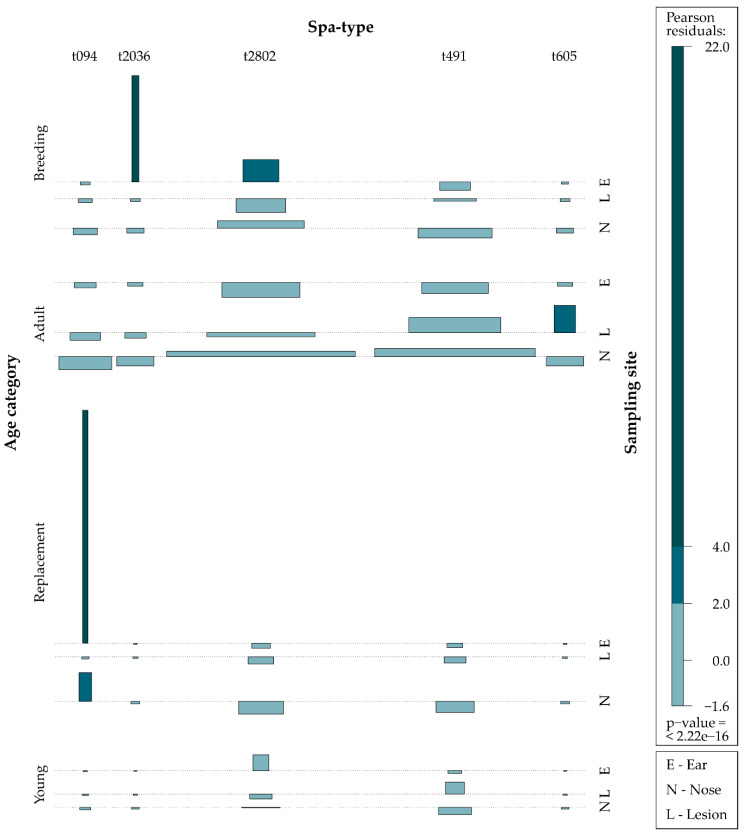
Association between spa-types, sampling site and age category: The association plot shows Pearson residuals comparing age category, spa-type and sampling site. Areas of the rectangles represent the number of observations in each cell. The darker the blue, the stronger the association.

**Table 1 animals-10-00774-t001:** Sampling distribution and prevalence of *Staphylococcus aureus*: *S. aureus* frequencies by age animal category, healthy anatomic site, and lesions.

Rabbit Age Category	No. of Animals (%)	*S. aureus* Frequency						
		Healthy Anatomic Sites						Lesions
		Nose No. (%)	Ear No (%)	Axilla No. (%)	Groin No. (%)	Perineum No. (%)	Total Isolates (%)	Isolates per Total of Lesions No. (%)
Young	26 (6)	4 (15.4)	9 (34.6)	3 (11.5)	4 (15.4)	6 (23.0)	26 (4.7)	1/5 (20)
Adult	312 (78)	139 (31.2)	135 (30.3)	50 (11.3)	62 (13.9)	59 (13.3)	445 (80.6)	33/39 (84.6)
Breeding	27 (7)	15 (25.9)	18 (31.0)	8 (13.8)	8 (13.8)	9 (15.5)	58 (10.8)	6/20 (30.0)
Replacement	35 (9)	3 (13.0)	12 (52.3)	2 (8.7)	3 (13.0)	3 (13.0)	23 (3.9)	0/2 (0.0)
Total	400	161 (29.2)	174 (31.5)	63 (11.4)	77 (13.9)	77 (13.9)	552 (93.2)	40/66 (60.6)

**Table 2 animals-10-00774-t002:** Risk of having lesions due to *S. aureus*: Rabbits stratified by number of anatomical sites colonized by *S. aureus*.

No. of Sites Colonized by *S. aureus*	Rabbits with Lesions by *S. aureus* (%)	Crude Analysis		Controlling for Age Category	
		OR	95% CI	OR *	95% CI *
0	2/116 (1.7)	1		1	
1	8/110 (7.3)	4.5	0.9–21.5	4.5	0.9–21.9
2	12/105 (11.4)	7.4	1.6–33.7	6.1	1.3–28.1
3	9/49 (18.4)	12.8	2.6–61.9	10.4	2.1–50.8
4	3/15 (20.0)	14.3	2.2–93.9	10.2	1.5–69.4
5	1/5 (20.0)	14.3	1.1–191.7	11.9	0.9–160.8

OR: Odds Ratio; CI: Confidential Interval; * association measure controlled for age category.

**Table 3 animals-10-00774-t003:** Features of clonal lineages: spa-types, spa-repetitions, spa-CCs, frequencies of spa-types, relative prevalence and confidence intervals.

Spa-Type	Repetitions	Spa-CC	Frequency (No)	Prevalence (%)	95% CI
t094	r07-r23-r12-r34-r34-r12-r12-r23	084	4	4.2	0.2–8.2
t491	r26-r23-r12-r34-r34-r12-r12-r23-r02-r12-r23	084	37	38.5	28.8–48.3
t605	r07-r23	-	2	2.1	0–4.9
t2036	r26-r23-r12-r34-r34-r12-r23-r02-r12-r23	012	2	2.1	0–4.9
t2802	r07-r23-r21-r17-r34-r34-r34-r33-r34	267	51	53.1	43.1–63.1
Total			96	100	

t: *spa*-type number; r: repetition number; spa-CC: *spa* Clonal Complexes; CI: Confidential Interval.

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
