# Peer review of "The Effect of Age and Sampling Site on the Outcome of Staphylococcus aureus Infection in a Rabbit (Oryctolagus cuniculus) Farm in Italy"

_animals, 2020, doi:10.3390/ani10050774_

Round 1
Reviewer 1 Report
Dear Authors,
Although the study is limited to a single farm, this manuscript is reasonably well presented and it contributes to greater understanding the main risk factors that could increase infections by S. aureus in rabitt farms. Overall I feel the manuscript would be of interest to the readers, particularly as it examines the association between spa-types, sampling site and age category with the prevalence of rabbits colonized by S. aureus in at least one body site, the nose area. In addition, the authors have utilised regression analysis to identify potential risk factors and confounders, and to predict expected prevalence of lesions associated with Staphylococcus spp on farm. The case is methodically and thoroughly well described.
However some minor lenguage editing would improve the readability.
In particolar the authors should be encouraged to ameliorate the abstract and results section in order to clearly improve the readability.
A few specific points are outlined below.
- In the title I would like to suggest to the authors to consider “sampling site” instead of “sample site”
- The authors report the isolation of three S. aureus strains from farm workers, after obtaining their informed consent, admitting that S. aureus strains, with same spa-type, could easily circulate in a community. For this finding of interest, I suggest to consider a fourth point in the aim (L72-74) concerning this circulation of S. aureus strains and its importance for the workers-animal interaction.
- L 84-95: in Materials and Methods, Animals and sampling section, please specify the cages condition (clear, dirty or rusted); This is important if you consider the environmental (and cage in this case) role in microorganism diffusion and persistence. Considering that S. aureus strains, both HV and LV, could came from environment, did the authors consider any cage or any environmental sampling? Did the authors evaluate the environmental contamonation of S. aureus with swabs or ather methods? If yes please add it to the text.
- L 106-108 and 116-117: these sentences could be reformulated more clearly for the reader.
- L 165-166: Please could you include the percentage with the numbers to better clarify each value?
- L 198-199: what do you mean with the sentence “but age category was not enough to explain the differences about the presence of lesions”? please clarify this point more clearly.
All the comments and suggestions I have made are of a minor nature and I would be happy for the manuscript to be accepted for publication once these have been addressed.
Hope my reviewing could help you to improve your work.
Kind Regards
Author Response
Dear Editor and Reviewers,
please see the attachment.
Best Regards,
Anna Rita Attili

Reviewer 2 Report
In this work titled “The effect of age and sample site on the outcome of Staphylococcus aureus infection in a rabbit (Oryctolagus cuniculus) farm in Italy”, Attili et al present the data of 403 animals where samples were taken from several locations such as ear, nose, axilla, groin, perineum and lesions. Authors aim to study the effect of age and site of infection, prevalence of lesions and genotypical characterization of S. aureus.
The overall work looks good for publication. However, authors could consider re writing the intro and results sections to make it easier to follow for a broader audience. The way is presented looks like only people in the field will keep attention to it.
Minor point:
Authors should consider including this reference: Viana et al. 2015. DOI:10.1038/ng.3219 to highlight the importance of the present study in understanding rabbit infections and zoonotic diseases
Author Response
Dear Editor and Reviewers,
please see the attachment.
Best regards,
Anna Rita Attili
